# Human airway cells prevent SARS-CoV-2 multibasic cleavage site cell culture adaptation

Mart M Lamers[1], Anna Z Mykytyn[1], Tim I Breugem[1], Yiquan Wang[2], Douglas C Wu[2†], Samra Riesebosch[1], Petra B van den Doel[1], Debby Schipper[1], Theo Bestebroer[1], Nicholas C Wu[2,3,4], Bart L Haagmans[1]*

[1]Viroscience Department, Erasmus Medical Center, Rotterdam, Netherlands; [2]Department of Biochemistry, University of Illinois at Urbana-Champaign, Urbana, United States; [3]Center for Biophysics and Quantitative Biology, University of Illinois at Urbana-Champaign, Urbana, United States; [4]Carl R. Woese Institute for Genomic Biology, University of Illinois at Urbana-Champaign, Urbana, United States

**Abstract** Virus propagation methods generally use transformed cell lines to grow viruses from clinical specimens, which may force viruses to rapidly adapt to cell culture conditions, a process facilitated by high viral mutation rates. Upon propagation in VeroE6 cells, SARS-CoV-2 may mutate or delete the multibasic cleavage site (MBCS) in the spike protein. Previously, we showed that the MBCS facilitates serine protease-mediated entry into human airway cells (Mykytyn et al., 2021). Here, we report that propagating SARS-CoV-2 on the human airway cell line Calu-3 – that expresses serine proteases – prevents cell culture adaptations in the MBCS and directly adjacent to the MBCS (S686G). Similar results were obtained using a human airway organoid-based culture system for SARS-CoV-2 propagation. Thus, in-depth knowledge on the biology of a virus can be used to establish methods to prevent cell culture adaptation.

*For correspondence:
b.haagmans@erasmusmc.nl

Present address: †Invitae Corporation, San Francisco, United States

Competing interests: The authors declare that no competing interests exist.

## Introduction

Severe acute respiratory syndrome coronavirus 2 (SARS-CoV-2) is the causative agent of the ongoing coronavirus disease (COVID-19) pandemic. SARS-CoV-2 emerged late 2019 in China and had spread globally within a few months (*Zhu et al., 2020*). An unprecedented rapid vaccine development response has led to approval of the first COVID-19 vaccines at the end of 2020. Conversely, the quest for efficacious specific antiviral therapies against SARS-CoV-2 was not successful. The lack of antivirals, the high adaptive capacity of the virus, and the emergence of new strains, indicate that further research on SARS-CoV-2 biology is necessary.

The first step in most SARS-CoV-2 laboratory studies is in vitro virus propagation to obtain highly concentrated virus stocks. Despite recent advances in physiologically relevant in vitro cell culture systems, methods to propagate clinical isolates have not changed since the first cell lines were established. Traditionally, virus propagation relies on transformed cell lines to produce progeny viruses after inoculation of these cells with a clinical specimen containing the virus. The most widely used cell line in virology is the Vero cell line, which is derived from the kidney of an African green monkey. This cell line and its derivatives (e.g. VeroE6, Vero118, etc) contain genomic deletions of genes involved in the antiviral interferon response (*Osada et al., 2014*). Such mutations are common in transformed cell lines and allow unbridled virus replication, facilitating the production of high titer virus stocks and allowing research on a wide range of viruses. These isolated viruses are often adapted to their cell culture systems due to their high mutation rates. The development of first and next generation sequencing methods revealed that these adapted viruses were heavily mutated and

had drifted significantly from their natural counterparts (*Alfson et al., 2018*; *Lau et al., 2020*; *Sutter and Moss, 1992*; *Tamura et al., 2013*; *Wei et al., 2017*). Cell culture adaptive mutations often affect viruses phenotypically, both in vitro and in vivo.

Coronavirus replication is initiated when the viral spike protein binds to the entry receptor on the cell and fuses viral and cellular membranes, allowing the viral RNA to enter the cytoplasm (*Hulswit et al., 2016*). The spike protein is composed of two domains, the S1 receptor binding domain and the S2 fusion domain, which are separated by the S1/S2 cleavage site. Proteolytic cleavage at the S1/S2 site and the more C-terminal S2' site is required for coronavirus infectivity as this turns on the fusogenic activity of the S2 domain (*Millet and Whittaker, 2015*). A distinctive feature of SARS-CoV-2 is the presence of a specific S1/S2 cleavage site in the viral spike protein (*Coutard et al., 2020*). The SARS-CoV-2 S1/S2 cleavage site contains three basic arginines interrupted by a non-polar alanine (RRAR) and is therefore referred to as a multibasic cleavage site (MBCS). This feature is remarkable as all other viruses within the clade of SARS-related viruses, including SARS-CoV, lack a PRRA insertion that creates this multibasic cleavage site, leading to speculations on whether this site is essential for efficient replication in the human respiratory tract (*Hoffmann et al., 2020*). Importantly, SARS-CoV-2 isolates that are cultured in the lab rapidly obtain mutations or deletions in the MBCS (*Davidson et al., 2020*; *Klimstra et al., 2020*; *Lau et al., 2020*; *Liu et al., 2020*; *Ogando et al., 2020*). On VeroE6 cells, these mutated viruses have a large plaque phenotype, grow to higher titers and outcompete the wild-type virus within 1–4 passages. These mutations have rarely been observed in human clinical specimens (*Liu et al., 2020*; *Wong et al., 2020*) and purified MBCS mutants do not efficiently replicate in hamsters (*Johnson et al., 2020*; *Lau et al., 2020*). We have recently shown that the MBCS is not required for entry into VeroE6 cells, but is essential for entry into human airway organoids (*Mykytyn et al., 2021*). We also reported that the MBCS facilitated plasma membrane serine protease-mediated entry, whereas it decreased the dependency on endosomal cathepsins for entry. The serine protease inhibitor camostat mesylate, but not a cathepsin inhibitor, effectively inhibited SARS-CoV-2 entry in human airway organoids, whereas the opposite was observed in VeroE6 cells. These findings demonstrate that SARS-CoV-2 enters relevant airway cells using serine proteases but not cathepsins, and suggest that the multibasic cleavage site is an adaptation to this viral entry strategy. The loss of the MBCS may be an adaptation to the cathepsin-mediated entry pathway present in VeroE6 cells.

In this study, we investigated whether mutations in the SARS-CoV-2 spike MBCS could be prevented in a human airway cell line (Calu-3) and 2D air-liquid interface (ALI) airway organoids in which SARS-CoV-2 enters using serine proteases.

## Results

SARS-CoV-2 isolates that are cultured in the lab rapidly lose their spike MBCS (*Davidson et al., 2020*; *Klimstra et al., 2020*; *Lau et al., 2020*; *Liu et al., 2020*; *Ogando et al., 2020*). To investigate the extent of cell culture adaptation in our SARS-CoV-2 stocks, we deep-sequenced passage 2, 3, and 4 stocks (P2, P3, and P4) of the BavPat-1 or Munich-1 strain propagated on VeroE6 cells. These stocks were produced from a P1 virus stock grown on VeroE6 cells (*Figure 1—figure supplement 1A*). In the P2 stock, the majority of reads (65.3%) in the MBCS were identical to the WT sequence (*Figure 1A*). In the multibasic RxxR motif both the first (R682L) and the last (R685H) arginine were mutated in 3.5% and 6.1% of reads, respectively. An additional mutation (S686G) directly C-terminal to the MBCS was detected at 25.1%. As this variant increased during passaging and therefore could be an adaptation to cell culture, we included it in our analyses. The P3 stock contained 18.8% wild-type (WT) viruses and the S686G was the major MBCS variant at 45.4% while mutations R685H and R682L were present at 22.4% and 7.3%, respectively. A deletion (Del679-688) of the entire MBCS was found as well at 6.1% (*Figure 1B*). In the P4 virus, the dominant variant contained the R685H mutation at 33.4%, while the Del679-688 and R682L increased to 13.9% and 10.4%, respectively. In the P4 virus, only 9% of reads were WT. Despite the strikingly low level of WT viruses in the VeroE6 stocks, the predominant cleavage motif for the P2 and P3 was still RRARS since mutations never co-occurred (*Figure 1A–C*). Our results show that a thorough analysis of deep-sequencing data is required to critically assess culture adaptation. The observations from deep-sequencing data were

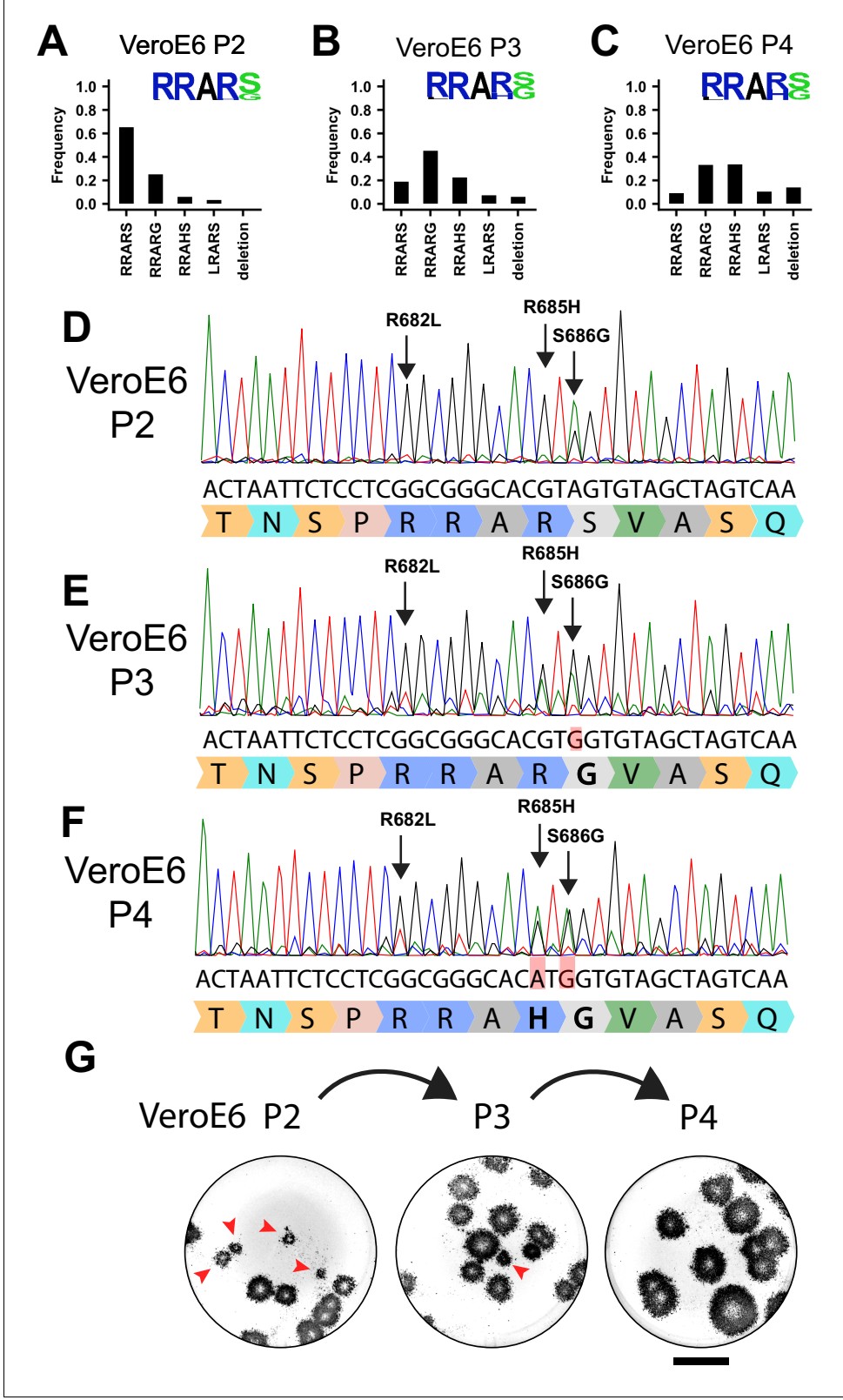

**Figure 1.** SARS-CoV-2 rapidly acquires multibasic cleavage site mutations when propagated on VeroE6 cells. (A–C) Deep-sequencing analysis of VeroE6 passage 2 (A), passage 3 (B), and passage 4 (C) virus stocks. In each graph, the amino acid sequence logo of the multibasic cleavage site is shown. (D–F) Sanger sequencing chromatograms of VeroE6 passage 2 (D), passage 3 (E), and passage 4 (F) viruses. Multibasic cleavage site mutations

*Figure 1 continued on next page*

*Figure 1 continued*

identified by deep-sequencing are indicated with arrows. Translated sequences are indicated below Sanger reads. (**G**) Plaque size analysis of VeroE6 passage 2–4 virus stocks on VeroE6 cells. Red arrow heads indicate small plaques. Scale bar indicates 1 cm.

The online version of this article includes the following figure supplement(s) for figure 1:

**Figure supplement 1.** Deep-sequencing analysis of VeroE6 passage 1 virus multibasic cleavage site and full genome deep-sequencing analysis of passage 1–4 viruses.

consistent with Sanger sequencing analysis (*Figure 1D–F*). In agreement with the mixed population of wildtype (WT) and mutant viruses, we observed small (non-adapted) and large (cell culture-adapted) plaque phenotypes in a plaque assay for the P2 virus, but plaques increased in size during passaging (*Figure 1G*).

Whereas the MBCS mutations directly removed arginines (R682L, R685H and the deletion) from the minimal RxxR furin motif, the most common stock mutation was the S686G. This site lies directly C-terminal from the MBCS at positions 682–685, indicating that it may also affect the MBCS functionally. To test this, we assessed the infectivity of the 686 mutation using vesicular stomatitis virus (VSV)-based pseudoviruses expressing a green fluorescent protein (GFP) as described before (*Mykytyn et al., 2021*). Western blot analysis of cleaved and uncleaved S1 revealed that proteolytic cleavage was observed for the WT SARS-CoV-2 pseudovirus and abrogated by all MBCS mutations tested (del-PRRA, del-RRAR, R682A, R685A, and R685H) (*Figure 2A,B*). For the S686G mutation ~10% cleaved S1 was observed, whereas this was ~80% for WT S (*Figure 2A,B*). The same difference in cleavage between WT and S686G pseudoviruses was observed for S2 (*Figure 2C,D*). As expected based on earlier work (*Mykytyn et al., 2021*), SARS-CoV-2 pseudoviruses with MBCS mutations were more infectious on VeroE6 cells and less infectious on Calu-3 cells (*Figure 3A–C*). A similar trend was observed for the S686G mutant spike. The infectivity on VeroE6-TMPRSS2 cells was similar for all spikes tested but the WT spike benefited more from TMPRSS2 expression (*Figure 3D–E*). Protease inhibitors camostat and E64D were then used to block serine proteases and cathepsins, respectively, to assess how spike mutations affect the route of entry. The stable expression of TMPRSS2 in VeroE6 cells leads to entry of WT pseudoviruses via this protease instead of cathepsin-mediated entry, but SARS-CoV-2 MBCS mutants and to a lesser extent the S686G mutant retained partial cathepsin mediated entry (*Figure 3F–I*). In addition, a GFP-complementation fusion assay, in which cell-cell fusion occurs at the plasma membrane, showed that MBCS mutations and to a lesser extent the S686G mutation abrogated fusion in VeroE6, VeroE6-TMPRSS2, and Calu-3 cells (*Figure 4A–C*). These data explain why VeroE6-propagated SARS-CoV-2 stocks rapidly accumulate mutations in the MBCS and at spike position 686. Despite being outside of the MBCS, the S686G mutation impairs spike cleavage, cell-cell fusion and serine protease usage, but not as dramatically as the MBCS mutations or deletions. The low infectivity of MBCS mutants and the S686G mutant on Calu-3 cells indicates that WT viruses could have a selective advantage in these cells.

In order to establish culture conditions in which SARS-CoV-2 is genetically stable, we tested whether WT viruses would have a selective advantage on Calu-3 cells that possess serine protease mediated entry and little cathepsin-mediated entry (*Mykytyn et al., 2021*). For these experiments, we first produced a Calu-3 P2 virus from the VeroE6 P1 stock (*Figure 5A*). This stock was 100% WT in the MBCS and no major variants (>50%) were detected in the rest of the genome (*Figure 5—figure supplement 1C*). An additional round of passaging on Calu-3 cells did not lead to any MBCS mutations, or mutations elsewhere (*Figure 5B*, *Figure 5—figure supplement 1C*). A Calu-3 P3 from a VeroE6 P2 virus did still contain the S686G at low frequency (7.4%) (*Figure 5C*), but continued passaging to P5 completely removed the S686G (*Figure 5D*). Again, we did not observe any other major variant mutations in the rest of the genome (*Figure 5—figure supplement 1C*). We also produced Calu-3 P4 virus from a VeroE6 P3 stock and we show that while this Calu-3 P4 virus had lost all MBCS mutations, the S686G mutation remained at a frequency of 65.7% (*Figure 5—figure supplement 1A*). The addition of E64D to block any cathepsin-mediated entry decreased the frequency of S686G by ~11%, to 54.3%, but did not remove S686G entirely (*Figure 5—figure supplement 1B*). These results support our earlier findings (*Figure 2*; *Figure 3*; *Figure 4*) that the S686G is a less severe cell culture adaptation compared with MBCS mutations, and more importantly show that Calu-3 cells can be used to grow genetically stable stocks without MBCS mutations or S686G.

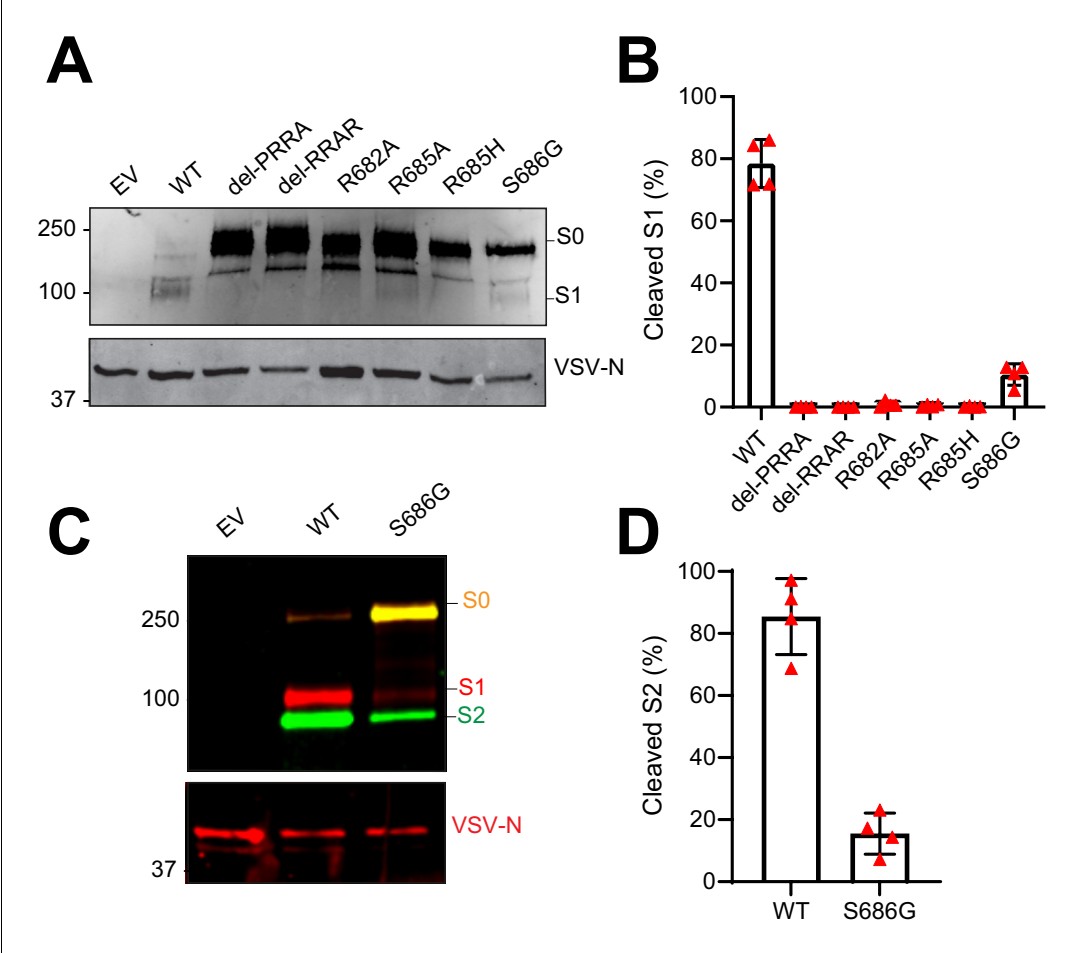

**Figure 2.** Mutations in the multibasic cleavage site and the adjacent serine residue (S686) abrogate S1/S2 cleavage. (A) Analysis of S1/S2 cleavage by S1 immunoblot of SARS-CoV-2 S (WT), multibasic cleavage site (MBCS) mutant and S686G mutant pseudoviruses. (B) Quantification of S1 cleavage from four independent pseudovirus productions. (C) Analysis of S1/S2 cleavage by multiplex S1 (red) and S2 (green) immunoblot of SARS-CoV-2 S (WT) and S686G mutant pseudoviruses. S0 indicates uncleaved spike; S1 indicates the S1 domain of cleaved spike; VSV-N indicates VSV nucleoprotein (production control). Numbers indicate the molecular weight (kDa) of bands of the protein standard. (D) Quantification of S2 cleavage from four independent pseudovirus productions. Error bars indicate SD. EV = empty vector. WT = wild type. kDa = kilo dalton.

Additionally, stocks grown on Calu-3 cells reached titers of $1.47 \times 10^6$–$2.1 \times 10^7$ TCID50/ml, indicating that Calu-3 cells support the production of high titer stocks.

To confirm that serine proteases are responsible for the reversal of cell culture adaptation observed in Calu-3 cells, we passaged the adapted VeroE6 P3 stock (*Figure 1B*) on regular VeroE6 cells or VeroE6-TMPRSS2 cells. P4 viruses grown on VeroE6 cells were only 9% WT and R685H was the dominant variant at 33.4% (*Figure 6A*; redisplay of *Figure 1C*). R682L and Del679-688 were present at 10.4% and 13.9%, respectively. Propagation of SARS-CoV-2 in VeroE6-TMPRSS2 cells resulted in an increase in the frequency of WT viruses at 21.7% and a decrease in the frequency of MBCS mutations (7.9% R685H; 4.2% R682L; 1.5% Del679-688), but the S686G remained at 64.6% (*Figure 6C*). The addition of the serine protease inhibitor camostat (10 μM) to the VeroE6-TMPRSS2 culture, but not the VeroE6 culture (*Figure 6B*), increased the frequency of MBCS mutations (36.6% R685H; 13% R682L; 8.8% Del679-688), confirming that serine proteases prevent cell culture adaptation (*Figure 6D*). As TMPRSS2 expression prevented MBCS mutations, we tested whether the addition of trypsin (0.7 μg/ml TPCK-Trypsin) would have a similar effect. Surprisingly, the addition of trypsin to VeroE6 cells, but not VeroE6-TMPRSS2 cells, led to deletion of the entire MBCS (*Figure 6—figure supplement 1A,C*). This deletion may arise due to the complete cleavage (S1/S2 and

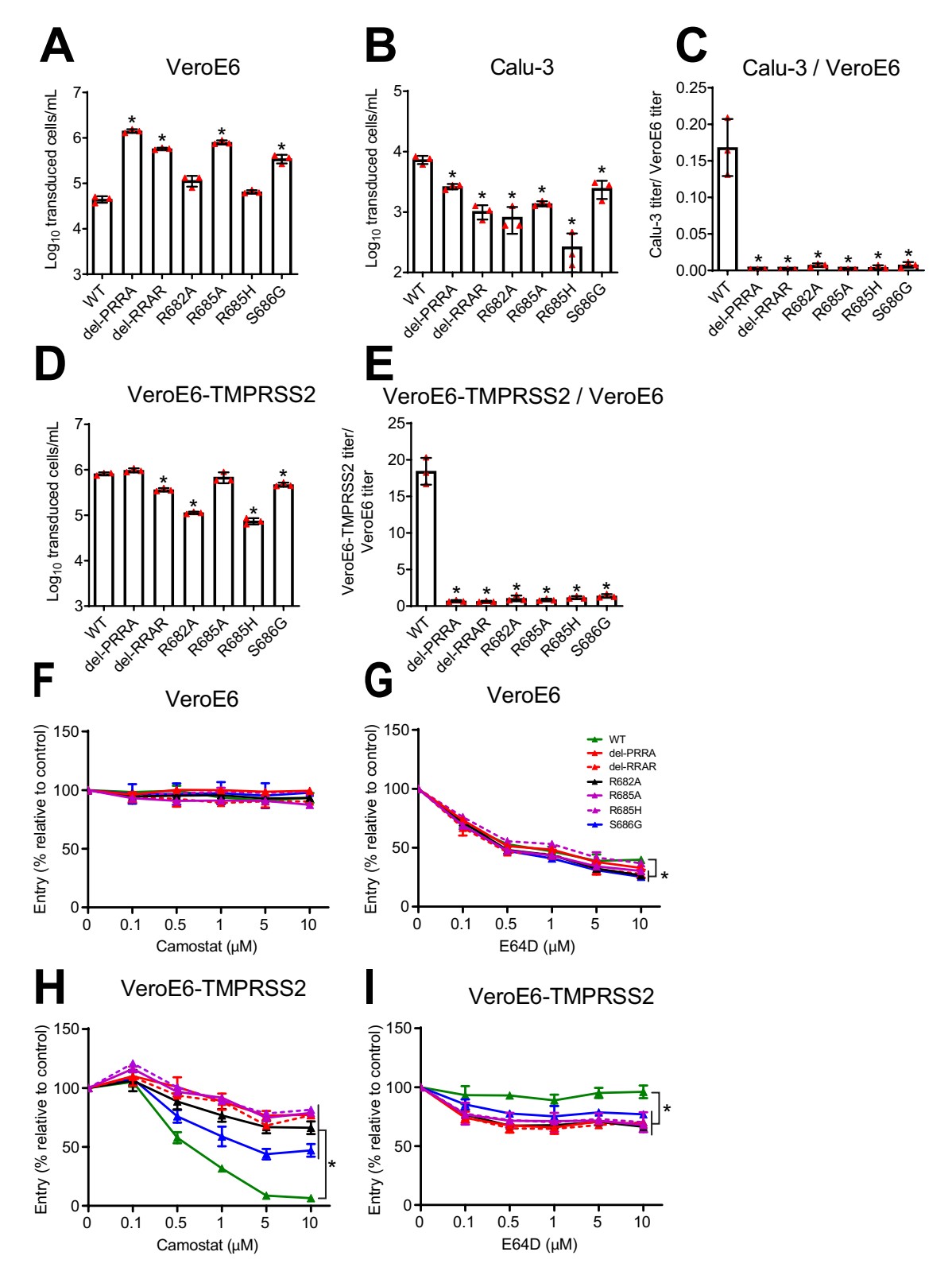

**Figure 3.** The SARS-CoV-2 multibasic cleavage site and the adjacent serine residue (S686) enhance infectivity and serine protease mediated entry on Calu-3 and VeroE6-TMPRSS2 cells. (A–B) SARS-CoV-2 (WT), multibasic cleavage site (MBCS) mutant and S686G pseudovirus infectious titers on (A) VeroE6 and (B) Calu-3 cells. (C) Fold change in SARS-CoV-2, MBCS mutant and S686G pseudovirus infectious titers on Calu-3 cells over infectious titers on VeroE6 cells. (D) SARS-CoV-2, MBCS mutant and S686G pseudovirus infectious titers on VeroE6-TMPRSS2 cells. (E) Fold change in SARS-CoV-2,

*Figure 3 continued on next page*

Figure 3 continued

MBCS mutant and S686G pseudovirus infectious titers on VeroE6-TMPRSS2 cells over infectious titers on VeroE6 cells. One-way ANOVA was performed for statistical analysis comparing all groups with WT. (F–I) SARS-CoV-2, MBCS mutant and S686G pseudovirus entry into (F and G) VeroE6 cells or (H and I) VeroE6-TMPRSS2 cells pre-treated with a concentration range of either (F and H) camostat mesylate or (G and I) E64D. Two-way ANOVA, followed by a bonferroni post hoc test was performed for statistical analysis comparing all groups to WT. WT pseudovirus entry into VeroE6 cells treated with 10 µM E64D was significantly different from del-RRAR, R682A, R685A and S686G pseudovirus entry. * indicates statistical significance (p<0.05) compared to WT (A–E). * indicates statistical significance (p<0.05) compared to WT at the highest inhibitor concentration (F–I). Experiments were performed in triplicate. Representative experiments from at least two independent experiments are shown. Error bars indicate SD. WT = wild type.

S2′) of virus particles that are not bound to the cellular membranes, which would inactivate them. Cell surface expressed TMPRSS2 could accelerate TMPRSS2-mediated entry and cell-cell spread, reducing the chance of trypsin cleavage in the supernatant. Additionally, we tested whether the addition of fetal bovine serum (FBS, heat-inactivated, 10% final concentration) affected culture adaptation as this is commonly added when producing viral stocks. FBS had a similar effect to trypsin in the VeroE6, but not the VeroE6-TMPRSS2 culture, indicating that proteases capable of cleaving spike may be present in serum and that FBS should be avoided when propagating SARS-CoV-2 (*Figure 6—figure supplement 1B,D*).

Next, we hypothesized that the best way to prevent cell culture adaptation would be to propagate the virus in non-transformed human airway cells. Recent advances in stem cell biology have enabled the establishment of human organoid culture systems (*Katsura et al., 2020*; *Nikolić et al., 2017*; *Sachs et al., 2019*; *Salahudeen et al., 2020*; *Sato et al., 2009*; *Youk et al., 2020*). These organoid cultures consist of stem cells that self-renew, allowing prolonged passaging and expansion, but can also differentiate to mature cell types, such as ciliated cells, goblet cells, and club cells. Organoids of the airways, the alveoli, and the intestine have recently been used by us to study SARS-CoV-2 entry and pathogenesis (*Katsura et al., 2020*; *Lamers et al., 2020a*; *Lamers et al., 2020b*; *Mykytyn et al., 2021*; *Salahudeen et al., 2020*; *Youk et al., 2020*; *Zhou et al., 2020*). Pseudoviruses containing MBCS mutations infected human airway organoids poorly (*Mykytyn et al., 2021*), indicating that these mutations could be prevented by virus propagation in these cells. To produce stocks in human airway organoids, we differentiated the organoids at 2D in transwell inserts at air-liquid interface for 12 weeks as described before (*Figure 7A*; *Mykytyn et al., 2021*). Apical cells, including ciliated cells, in these cultures expressed TMPRSS2 as shown by immunohistochemistry (*Figure 7B*). To produce viral stocks, 2D airway organoids were inoculated at the apical side at a MOI of 0.05 using the VeroE6 P2 stock (*Figure 1A*). After a 2-hr incubation, cells were washed three

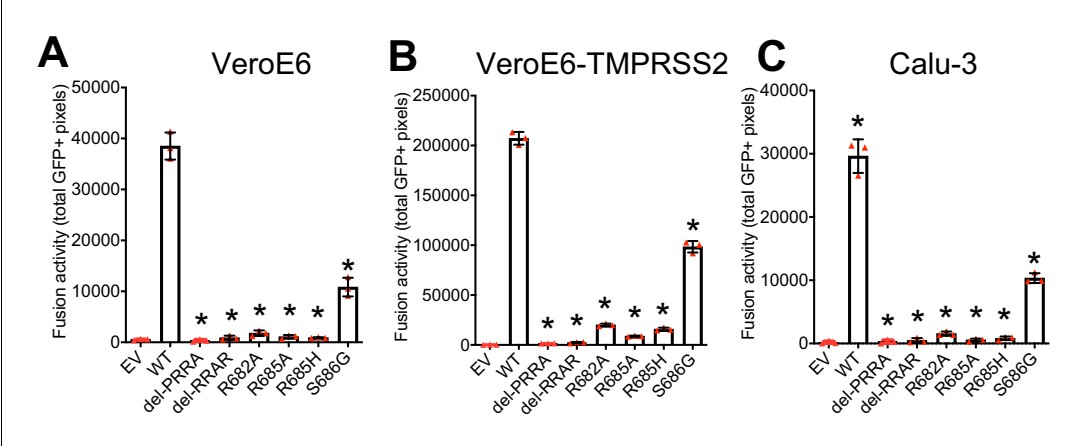

**Figure 4.** Multibasic cleavage site mutations and the adjacent serine residue (S686) impair spike protein fusogenicity. (A–C) Fusogenicity of wild type SARS-CoV-2 spike and spike mutants was assessed after 18 hr by measuring the sum of all GFP+ pixels per well in a GFP-complementation fusion assay on VeroE6-GFP1-10 (A), VeroE6-TMPRSS2-GFP1-10 (B), and Calu-3-GFP1-10 (C) cells. The experiment was performed in triplicate. A representative experiment from two independent experiments is shown. Statistical analysis was performed by one-way ANOVA. * indicates a significant difference compared to WT (p<0.05). Error bars indicate SD. EV = empty vector. WT = wild type.

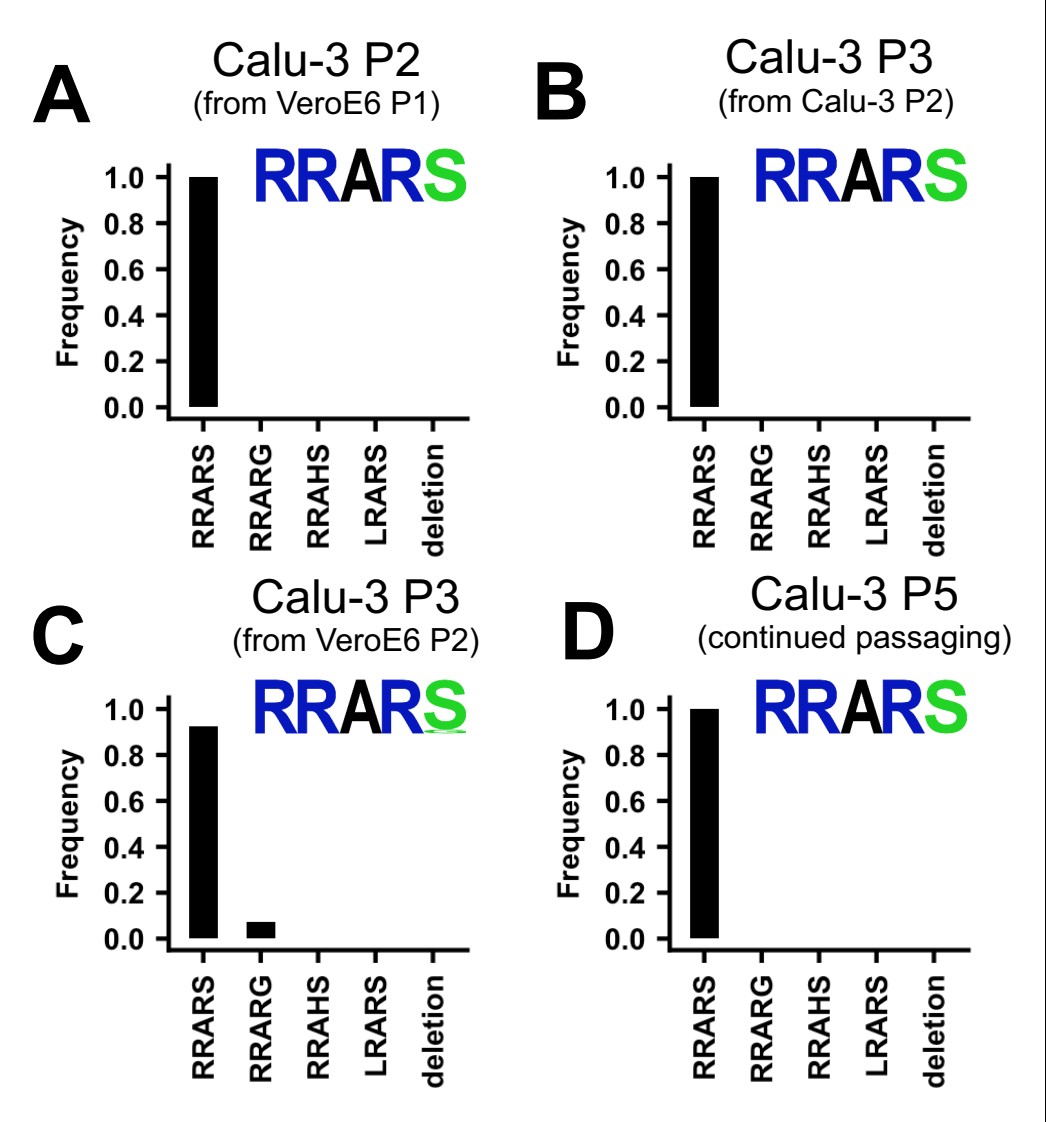

**Figure 5.** SARS-CoV-2 propagation in Calu-3 cells efficiently prevents SARS-CoV-2 cell culture adaptation. (**A**) Deep-sequencing analysis of Calu-3 passage 2 virus from a VeroE6 passage 1. (**B**) Deep-sequencing analysis of Calu-3 passage 3 virus from the Calu-3 passage 2 in A. (**C**) Deep-sequencing analysis of Calu-3 passage 3 virus grown from a VeroE6 passage 2 stock (*Figure 1A*). Deep-sequencing analysis of Calu-3 passage 5 virus from a Calu-3 passage 3 stock in C. In each graph, the amino acid sequence logo of the multibasic cleavage site is shown.

The online version of this article includes the following figure supplement(s) for figure 5:

**Figure supplement 1.** Multibasic cleavage site deep-sequencing analysis of passage 4 Calu-3 viruses from an adapted VeroE6 P3 stock and full genome deep-sequencing analysis of Calu-3-propagated viruses.

times to remove unbound particles. On day 2–5 post-infection, apical washes were collected and stored at 4°C. During virus collections, bound virus particles were released from cells by pipetting directly on the cell layer. Virus collections from day 2 and day 3 (d2+3), and day 4 and day 5 (d4+5) were pooled, centrifuged, and filtered to remove debris, dead cells and mucus. In these cultures, ciliated cells were infected, as shown by confocal imaging at day 3 post infection (*Figure 7C*). At day 5, cultures exhibited widespread infection (*Figure 7D*) and significant cytopathic effects including loss of ciliated cells (*Figure 7D–E*) and syncytium formation (*Figure 7E*). To remove cytokines that could interfere in downstream experiments (such as interferons), we exchanged the medium in the filtered virus collections three times using an Amicon Ultra-15 column (100 kDa cutoff). The resulting

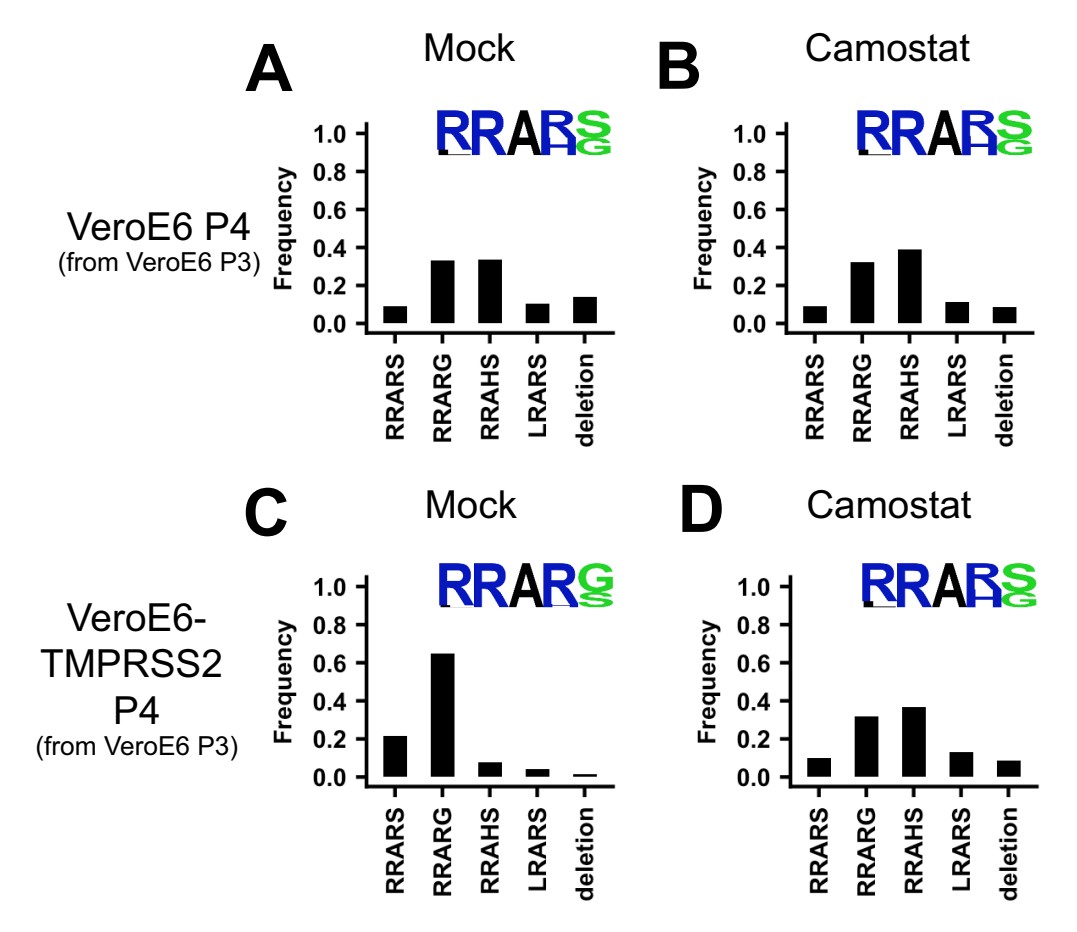

**Figure 6.** Serine protease expression prevents MBCS mutations. (A–B) Deep-sequencing analysis of VeroE6 passage 4 virus from a VeroE6 passage 3 (A is a redisplay of *Figure 1C*) mock-treated or treated with 10 µM camostat. (C–D) Deep-sequencing analysis of VeroE6-TMPRSS2 passage 4 virus from a VeroE6 passage 3 mock-treated or treated with 10 µM camostat. In each graph the amino acid sequence logo of the multibasic cleavage site is shown. The online version of this article includes the following figure supplement(s) for figure 6:

**Figure supplement 1.** Multibasic cleavage site and full genome deep-sequencing analysis of passage 4 VeroE6 and VeroE6-TMPRSS2 viruses.

titers from the d2+3 and d4+5 stocks were $5.64 \times 10^5$ and $1.00 \times 10^7$ TCID$_{50}$/ml, respectively, indicating that high titer virus stocks can be made in human airway organoids. Sequencing demonstrated that the high titer organoid stock (d4+5) had a 98.9% WT spike sequence, without multibasic cleavage site mutations and the S686G mutation at only 1.1% (*Figure 8A–B*). In accordance, the Organoid P3 virus produced small plaques (*Figure 8C*). No major variants were detected in the rest of the genome (*Figure 8D*). Next, we investigated S1/S2 cleavage of the VeroE6 P2, P3, Calu-3 P3, and Organoid P3 virus stocks by immunoblot (*Figure 8E*). The non-adapted Calu-3 and organoid stocks were >85% cleaved, while the VeroE6 P2 and P3 stocks were 71.2% and 33% cleaved, respectively (*Figure 8F*). The findings support that the Calu-3 and organoid stocks are non-adapted and indicate that in vivo the S1/S2 cleavage takes place in the producing cell.

## Discussion

The rapid loss of the SARS-CoV-2 MBCS in cell culture has underlined that some in vitro propagation systems may fail to model key aspects of the viral life cycle. As these mutations directly affect the relevance and translatability of all laboratory SARS-CoV-2 experiments, it is pivotal to sort out exactly why these occur in order to prevent them. We and others have previously reported that the SARS-CoV-2 MBCS enhances serine protease-mediated entry, the dominant entry pathway in human

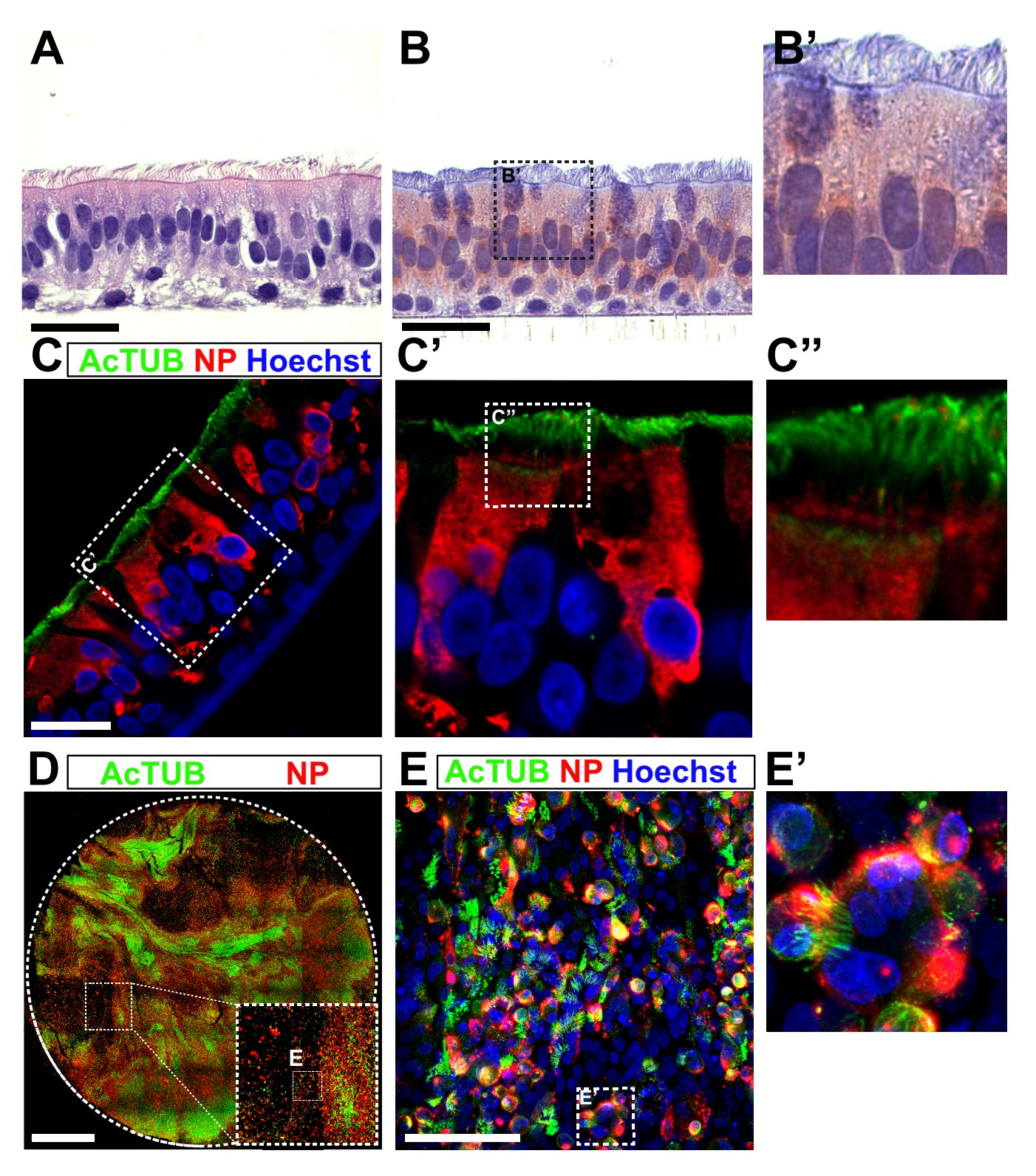

**Figure 7.** A 2D air-liquid interface human airway organoid model for SARS-CoV-2 propagation. (**A**) Human airway organoids were dissociated and plated onto 12 mm transwell inserts. After an 8–12 week differentiation period at air-liquid interface cultures contained ciliated, non-ciliated and basal cells as shown on a hematoxylin-eosin stain. (**B**) Air-exposed cells, but not basal cells, expressed the priming protease TMPRSS2 as shown by immunohistochemistry. (**C**) Immunofluorescent staining indicated that in these cultures, ciliated cells (acetylated tubulin+ or AcTUB+ cells) were

*Figure 7 continued on next page*

*Figure 7 continued*

infected by SARS-CoV-2. (**D and E**) At 5 days post-infection, whole-well confocal imaging indicated the infection was widespread (**D**) and cytopathic effects, including cilia damage (**D and E**) and syncytial cells (**E**) were visible. Scale bars indicate 20 μm in A, B, C; 2 mm in D; and 100 μm in E.

airway cells (*Hoffmann et al., 2020*; *Mykytyn et al., 2021*). VeroE6 cells, commonly used in the lab to grow virus stocks, lack this entry pathway, forcing the virus to use endosomal cathepsins for entry. Cell culture adaptations promoting cathepsin-mediated entry were also observed for the human coronavirus 229E and adapted viruses showed a reduced ability to replicate in differentiated airway epithelial cells (*Bertram et al., 2013*; *Shirato et al., 2017*). These observations led us to hypothesize that mutations in the MBCS could be prevented in cells with an active serine protease-mediated entry pathway. In this study, we show that the ectopic expression of the serine protease TMPRSS2 in VeroE6 cells prevented MBCS mutations. Virus propagation in Calu-3 cells, which naturally express serine proteases, also prevented cell culture adaptation. Similar results were obtained using a human airway organoid-based culture system for SARS-CoV-2 propagation.

Our study shows that SARS-CoV-2 rapidly adapts to VeroE6 cell culture. Therefore, deep-sequencing of viral stocks, which offers a thorough analysis beyond the consensus sequence, is essential. As none of the MBCS mutations co-occurred, consensus sequence logos of culture adapted stocks were often WT, while actually only 10–20% of viral reads contained the WT sequence. Therefore, besides reporting the consensus sequence SARS-CoV-2 studies should prefera-bly also report the percentage of WT reads in the MBCS. The first adaptation to occur in our stocks was the S686G mutation, which lies directly adjacent to the MBCS and decreased Calu-3 infectivity, fusogenicity and S1/S2 cleavage, but not as severely as MBCS mutations. Interestingly, this mutation is rapidly positively selected in ferrets (*Richard et al., 2020*), and also transmitted, suggesting that there are key differences in transmission between humans and ferrets. Alternatively, it is possible that S686G optimizes cleavage by a specific ferret protease.

SARS-CoV-2 is generally grown on VeroE6 cells in the presence of 1–10% FBS, as this allows the production of highly concentrated virus stocks. Here, we show that SARS-CoV-2 rapidly acquired mutations in the MBCS upon passaging in VeroE6 cells and that the addition of FBS increases the frequency of MBCS mutations. Currently, we do not know which components of FBS lead to the increased rate of cell culture adaptation, but we hypothesize that (serine) proteases, naturally pres-ent in serum (*Shimomura et al., 1992*) may be responsible as the addition of trypsin dramatically increased the frequency of MBCS mutations and even led to the deletion of the entire MBCS. The observation that the MBCS is a disadvantage in the presence of trypsin indicates that S2' proteolytic cleavage (performed by trypsin *Hoffmann et al., 2020*) should not occur in the supernatant where it would cause all spikes to shed their S1 and adopt their post fusion conformation prior to encounter-ing the plasma membrane. Alternatively, protease inhibitors in serum (*Gstraunthaler, 2003*) may block transmembrane serine proteases. For these reasons, the use of serum should be avoided when producing virus stocks. The use of defined serum-free media avoids the uncertainty that fac-tors in serum affect the genetic stability of a virus and increase experimental reproducibility due to variations in serum sources.

We show that the expression of the serine protease TMPRSS2 decreases the replicative fitness of MBCS mutant SARS-CoV-2 viruses, which can then be outcompeted by WT viruses. This indicates that the MBCS is an adaptation to serine proteases and that the serine protease-mediated entry pathway is used for entry in vivo. This is in agreement with our earlier observations that SARS-CoV-2 enters using serine proteases on airway organoids (*Mykytyn et al., 2021*) and that MBCS mutant pseudoviruses could not efficiently infect these cells. Low infectivity of MBCS mutants on the airway cell line Calu-3 was also noted by *Hoffmann et al., 2020*. In contrast, two CRISPR-based survival screens recently identified several endosomal proteins, including cathepsin L, as essential SARS-CoV-2 genes (*Daniloski et al., 2021*; *Wei et al., 2020*). As noted recently by *Bailey and Diamond, 2021*, the identification of endosomal host factors as proviral in cell-line-based CRISPR screens requires validation in primary cells. Our observation that WT viruses have a selective advantage in 2D airway organoids confirms that the endosomal entry pathway is of little significance in relevant cells.

As new SARS-CoV-2 strains are emerging now and will continue to emerge for as long as SARS-CoV-2 circulates in humans, there is a need to develop propagation systems that will preserve

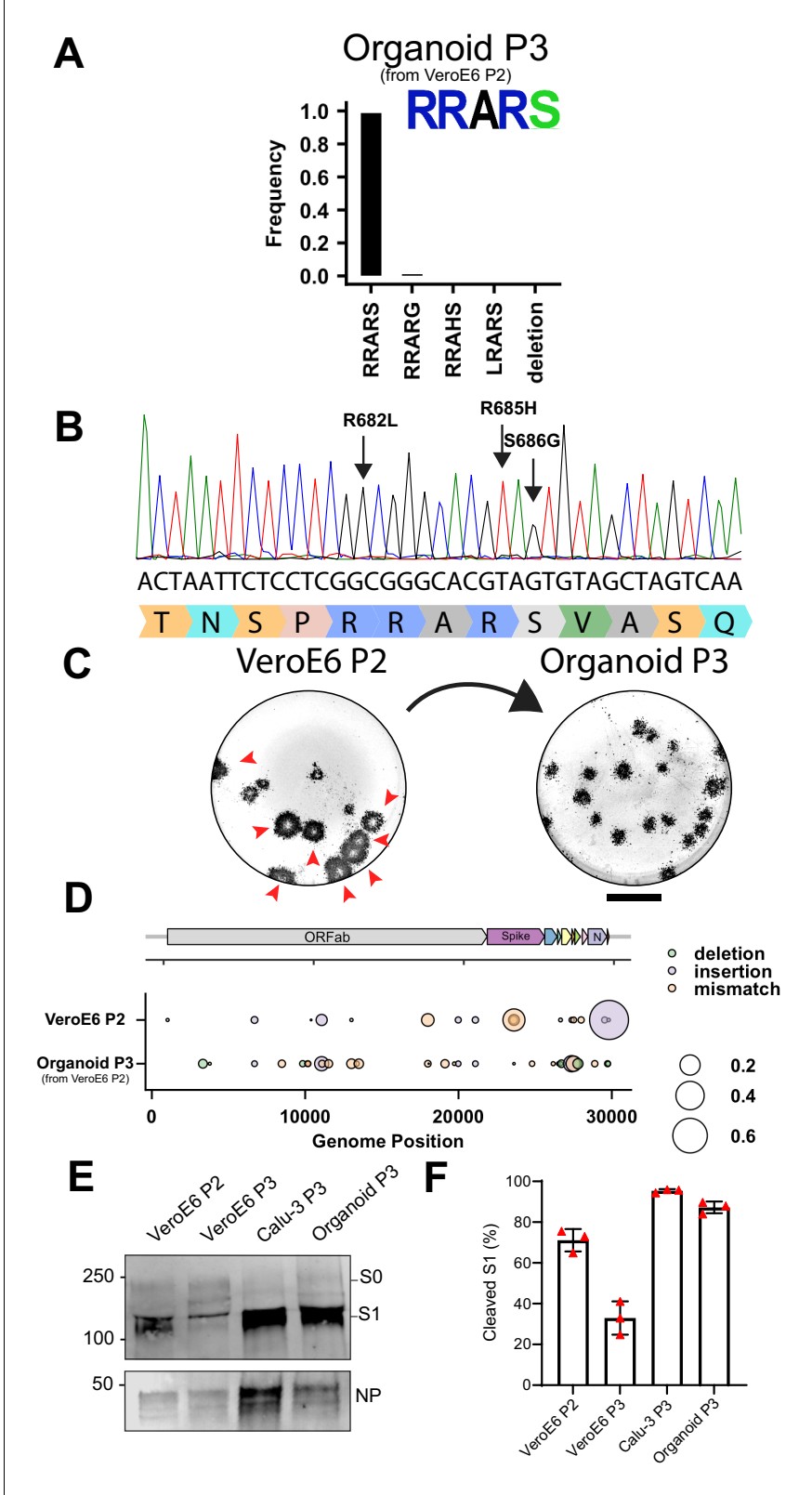

**Figure 8.** 2D air-liquid interface human airway organoids produce high titer stocks without multibasic cleavage site mutations. (A–B) Deep-sequencing analysis (A) and Sanger chromatogram (B) of Organoid passage 3 virus from a VeroE6 passage 2 stock (*Figure 1A*). The amino acid sequence logo of the multibasic cleavage site is shown. The translated sequence is indicated below the Sanger read. Arrows indicate where cell culture adaptations to VeroE6 cells occur. (C) Plaque size analysis of VeroE6 passage 2 and Organoid passage 3 virus (the VeroE6 data is a redisplay of *Figure 1G*). Red arrow

*Figure 8 continued on next page*

*Figure 8 continued*

heads indicate large plaques. Scale bar indicates 1 cm. (D) Full genome deep-sequencing analysis of VeroE6 passage 2 and organoid passage three stocks. In D VeroE6 P2 is a redisplay of VeroE6 P2 in *Figure 1—figure supplement 1B*. (E) Immunoblot analysis of VeroE6 passage 2 and 3, Calu-3 passage 3 and Organoid passage 3 stocks. S0 indicates uncleaved spike; S1 indicates the S1 domain of cleaved spike; NP indicates nucleoprotein. Numbers indicate the molecular weight (kDa) of bands of the protein standard. (F) Quantification of cleavage from three immunoblots. Error bars indicate SD.

The online version of this article includes the following figure supplement(s) for figure 8:

**Figure supplement 1.** Schematic workflow for the production of SARS-CoV-2 stocks on 2D air-liquid interface differentiated airway organoids.

genetic stability for any given SARS-CoV-2 mutant originating from a human respiratory sample. The human airway organoid model for SARS-CoV-2 propagation established in this study (*Figure 8*) allows high titer SARS-CoV-2 production and was most successful in removing MBCS mutations. We have not observed any major variants associated with virus propagation in this system, but we cannot exclude that some minor variants could be culture adaptations. The self-renewing capacity of organoids allows labs to share organoid lines, allowing a level of reproducibility similar to that of transformed cell lines. In addition, organoids can be grown from a wide range of organs and species to best model the in vivo environment of a particular virus. As an accurate modeling of viral target cells is likely to prevent most forms of cell culture adaptation, we expect that in the future organoid-based systems are likely to replace transformed cell lines for viral stock production.

In conclusion, this study shows that SARS-CoV-2 rapidly adapts to VeroE6 cell culture propagation and that this can be prevented by using cell lines with an active serine protease-mediated entry pathway (VeroE6-TMPRSS2 or Calu-3). Alternatively, a 2D airway organoid-based cell culture model can be used for SARS-CoV-2 propagation if in the future new variants emerge that are not genetically stable on Calu-3 cells. Our study also shows that deep-sequencing rather than consensus sequencing of viral stocks is critical for obtaining relevant and reproducible results in SARS-CoV-2 studies.

# Materials and methods

**Key resources table**

| Reagent type (species) or resource | Designation | Source or reference | Identifiers | Additional information |
|---|---|---|---|---|
| Antibody | Rabbit-anti-SARS-CoV NP (polyclonal) | Sino Biological | Cat# 40143-T62 | IF (1:1000) |
| Antibody | Mouse anti-TMPRSS2 (monoclonal) | Santa Cruz | Cat# sc-515727 | IHC (1:200) |
| Antibody | Goat-anti-mouse | Dako | Cat# P0260 | IF (1:400) |
| Antibody | Goat anti-rabbit IgG (H+L) Alexa Fluor Plus 594 | Invitrogen | Cat# A32740 | IF (1:400) |
| Antibody | Goat anti-mouse IgG (H+L) Alexa Fluor 488 | Invitrogen | Cat# A11029 | IF (1:2000) |
| Antibody | Mouse-anti-AcTub IgG2A Alexa Fluor 488 (monoclonal) | Santa Cruz Biotechnology | Cat# sc-23950 AF488 | IF (1:100) |
| Antibody | Mouse anti-nucleocapsid | Sinobiological | Cat# 40143-MM05 | IF (1:1000) |
| Antibody | Rabbit anti-SARS-CoV S1 (polyclonal) | Sinobiological | Cat# 40150-T62 | WB (1:1000) |
| Antibody | Mouse-anti-SARS-CoV-2 S2 (monoclonal) | Genetex | Cat# GTX632604 | WB (1:1000) |
| Antibody | Mouse-anti-VSV-N (monoclonal) | Absolute Antibody | Cat# Ab01403-2.0 | WB (1:1000) |
| Biological sample (*Homo sapiens*) | Airway organoids | *Mykytyn et al., 2021* | | |

*Continued on next page*

*Continued*

| Reagent type (species) or resource | Designation | Source or reference | Identifiers | Additional information |
|---|---|---|---|---|
| Cell line (*Cercopithecus aethiops*) | VeroE6 | ATCC | CRL 1586TM | |
| Cell line (*Cercopithecus aethiops*) | VeroE6 TMPRSS2 | *Mykytyn et al., 2021* | | |
| Cell line (*Cercopithecus aethiops*) | VeroE6 GFP1-10 | *Mykytyn et al., 2021* | | |
| Cell line (*Cercopithecus aethiops*) | VeroE6 GFP1-10 TMPRSS2 | *Mykytyn et al., 2021* | | |
| Cell line (*Homo sapiens*) | Calu-3 | ATCC | HTB 55 | |
| Cell line (*Homo sapiens*) | Calu-3 GFP1-10 | *Mykytyn et al., 2021* | | |
| Chemical compound, drug | E64D | MedChemExpress | Cat# HY-100229 | |
| Chemical compound, drug | Camostat mesylate | Sigma | Cat# SML0057 | |
| Chemical compound, drug | Polyethylenimine linear | Polysciences | Cat# 23966 | |
| Chemical compound, drug | Hygromycin B | Invitrogen | Cat# 10843555001 | |
| Chemical compound, drug | Geneticin | Invitrogen | Cat# 10131035 | |
| Chemical compound, drug | Avicel | FMC biopolymers | - | |
| Chemical compound, drug | Laemmli | BioRad | Cat# 1610747 | |
| Commercial assay or kit | SuperScript IV Reverse Transcriptase | Invitrogen | Cat# 18090200 | |
| Commercial assay or kit | Pfu Ultra II Fusion HS DNA Polymerase | Agilent Technologies | Cat# 600674 | |
| Commercial assay or kit | Qiaquick PCR Purification Kit | QIAGEN | Cat# 28104 | |
| Commercial assay or kit | BigDye Terminator v3.1 Cycle Sequencing Kit | Applied Biosystems | Cat# 4337456 | |
| Commercial assay or kit | ProtoScript II Reverse Transcriptase | New England BioLabs | Cat# NEB M0368X | |
| Commercial assay or kit | KAPA HyperPlus | Roche | Cat# 7962428001 | |
| Other | Amicon Ultra-15 Centrifugal Filter Unit with Ultracel-100 membrane | Millipore | Cat# UFC910024 | |
| Other | Opti-MEM I (1X) + GlutaMAX | Gibco | Cat# 51985–042 | |
| Other | Advanced DMEM/F12 | Thermo Fisher scientific | Cat# 12634–010 | |
| Other | AO medium | *Sachs et al., 2019* | N/A | |
| Other | Pneumacult ALI medium | Stemcell | Cat # 05001 | |
| Other | TryplE | Thermo Fisher scientific | Cat# 12605010 | |
| Other | Basement membrane extract | R and D Systems | Cat# 3533-005-02 | |
| Other | Transwell inserts | Corning | Cat# 3460 | |
| Other | Collagen Type I, High concentration Rat tail | Corning | Cat# 354249 | |

*Continued on next page*

*Continued*

| Reagent type (species) or resource | Designation | Source or reference | Identifiers | Additional information |
|---|---|---|---|---|
| Other | 0.45 µm low protein binding filter | Millipore | Cat# SLHV033RS | |
| Other | Hoechst | Thermo Fisher | Cat# H1399 | |
| Other | Ampure XP Beads | Beckman Coulter | Cat# A63882 | |
| Other | Illumina sequencer V3 MiSeq flowcell | Illumina | | |
| Other | ABI PRISM 3100 Genetic Analyzer | Applied Biosystems | | |
| Other | Odyssey CLx | Licor | | |
| Other | Amersham Typhoon Biomolecular Image | GE Healthcare | | |
| Other | Amersham Imager 600 | GE Healthcare | | |
| Other | LSM700 confocal microscope | Zeiss | | |
| Other | Carl ZEISS Vert.A1 | Zeiss | | |
| Software, algorithm | ZEN | Zeiss | | |
| Software, algorithm | ImageQuant TL 8.2 | GE Healthcare | | |
| Software, algorithm | Studio Lite Ver 5.2 | Licor | | |
| Software, algorithm | GraphPad PRISM 8, 9 | GraphPad | | |
| Software, algorithm | Illustrator | Adobe inc | | |
| Strain, strain background (SARS-CoV-2) | SARS-CoV-2 BavPat-1 | Dr. Christian Drosten | European Virus Archive Global #026 V-03883 | |

## Cell lines

VeroE6 wildtype and retrovirally transduced cell lines were maintained in Dulbecco's modified Eagle's medium (DMEM, Lonza) supplemented with 10% fetal bovine serum (FBS, Sigma, F7524, heat inactivated for 30 min at 56℃), HEPES, sodium bicabonate, penicillin (100 IU/mL), and strepto-mycin (100 IU/mL). VeroE6-TMPRSS2, VeroE6-GFP1-10, VeroE6-TMPRSS2-GFP1-10, and Calu-3-GFP1-10 cells were generated as described before (*Mykytyn et al., 2021*). Calu-3 and Calu-3-GFP1-10 cells were maintained in Eagle's modified Eagle's medium (EMEM) supplemented with 10% FBS, penicillin (100 IU/mL) and streptomycin (100 IU/mL). All cell lines were grown at 37℃ in a humidified CO2 incubator, and transduced cell lines were cultured in the presence of selection antibiotics. VeroE6 and Calu-3 cells were derived from ATCC. They were tested negative for mycoplasma.

## SARS-CoV-2 propagation in cell lines

SARS-CoV-2 (isolate BetaCoV/Munich/BavPat1/2020; European Virus Archive Global #026 V-03883; kindly provided by Dr. C. Drosten) was propagated to the indicated passage on VeroE6, VeroE6-TMPRSS2 or Calu-3 cells, as indicated, in Advanced DMEM/F12 (Gibco), supplemented with HEPES, Glutamax, penicillin (100 IU/mL) and streptomycin (100 IU/mL) (AdDF+++) at 37℃ in a humidified CO2 incubator. Infections were performed at a multiplicity of infection (MOI) of 0.01 and virus was harvested after 72 hr. The culture supernatant was cleared by centrifugation and stored at −80℃. Calu-3 stocks were additionally cleared using a 0.45 µM low protein binding filter (Millipore) to remove mucus debris produced by these cells and the medium was exchanged three times for Opti-MEM I (1X) + GlutaMAX (Gibco) using an Amicon Ultra-15 column (100 kDa cutoff). At the end of each centrifugation step, approximately 2 ml was left in the top compartment. After three exchanges, the purified virus was transferred to a new 50 ml tube and the Amicon Ultra-15 column was washed ten times by adding 1 ml Opti-MEM I (1X) + GlutaMAX (Gibco) to the top compart-ment, pipetting up and down several times on the filter and adding each wash to the tube contain-ing the purified virus preparation. This step was repeated until the volume in the purified virus stock

was equal to the original volume of culture supernatant. Purified virus was stored at −80°C in aliquots. Stock titers were determined by preparing 10-fold serial dilutions in Opti-MEM I (1X) + GlutaMAX (Gibco). One-hundred μl of each dilution was added to monolayers of $2 \times 10^4$ VeroE6 cells in the same medium in a 96-well plate. Plates were incubated at 37°C for 5 days and then examined for cytopathic effect. The TCID50 was calculated according to the method of Spearman and Kärber. All work with infectious SARS-CoV-2 was performed in a Class II Biosafety Cabinet under BSL-3 conditions at Erasmus Medical Center.

## Cloning

Cloning of SARS-CoV-2 S WT, del-PRRA, R685A, and R685H constructs for pseudovirus production and GFP-complementation fusion assay was performed as described before (*Mykytyn et al., 2021*). Del-RRAR, R682A, and S686G plasmids were generated by mutagenesis PCR.

## Organoid culture and differentiation

Human airway stem cells were isolated and grown into organoids, and passaged as described before (*Lamers et al., 2020a*) using a protocol adapted from *Sachs et al., 2019*. Adult lung tissue was obtained from residual, tumor-free, material obtained at lung resection surgery for lung cancer. The Medical Ethical Committee of the Erasmus MC Rotterdam granted permission for this study (METC 2012–512). Study procedures were performed according to the Declaration of Helsinki, and in compliance with relevant Dutch laws and institutional guidelines. The tissues obtained were anonymized and non-traceable to the donor. In this study we used organoids from one donor, from which bronchial and bronchiolar organoids were grown. Differentiation of human airway organoids at air-liquid interface was performed as described before (*Lamers et al., 2020a*). Cultures were differentiated for 8–12 weeks at air-liquid interface.

## SARS-CoV-2 stock production on 2D air-liquid interface human airway organoids

To produce stocks in human airway organoids, we differentiated the bronchial organoids in transwell inserts at air-liquid interface for twelve weeks. A total of 12 12 mm transwell inserts were washed three times in AdDF+++ before inoculation at the apical side at a MOI of 0.05. After a 2-hr incubation, cells were washed three times with AdDF+++ to remove unbound particles. Twenty-four hours post-infection, cells were washed by adding 500 ul AdDF+++ to the apical side of the cells and incubating at 37°C 5% $CO_2$ for 30 min to disperse the newly produced virus particles, facilitating the next round of infection. Next, the medium was removed and discarded, as generally little virus is produced in the first 24 hr (*Lamers et al., 2020a*). At days 2–5 post-infection, washes were collected and stored at 4°C. During collections, bound virus particles were removed from the cells by pipetting three times directly on the cell layer after the 30 min incubation step at 37°C 5% $CO_2$. Virus collections from day 2 and day 3 (d2+3) and day 4 and day 5 (d4+5) were pooled, mixed by pipetting, centrifuged at 4000 x g for 4 min, and filtered through a 0.45 um low protein binding filter (Millipore) to remove debris, dead cells, and mucus. To remove cytokines that could interfere in downstream experiments (such as interferons), we exchanged the medium in the filtered virus collections three times with Opti-MEM I (1X) + GlutaMAX (Gibco) using an Amicon Ultra-15 column (100 kDa cutoff). At the end of each centrifugation step, approximately 2 ml was left in the top compartment. After three exchanges, the purified virus was transferred to a new 50 ml tube and the Amicon Ultra-15 column was washed ten times by adding 1 ml Opti-MEM I (1X) + GlutaMAX (Gibco) to the top compartment, pipetting up and down several times on the filter and adding each wash to the tube containing the purified virus preparation, resulting in a total volume of ~12 ml. Next, virus preparations were aliquoted in 500 μl aliquots, stored at −80°C and thawed for titrations on VeroE6 cells.

## Pseudovirus assay

Pseudovirus production, infectivity, and entry assays were performed as described before (*Mykytyn et al., 2021*). Briefly, pseudoviruses expressing WT, MBCS mutant, and S686G S were titrated by preparing 10-fold serial dilutions in Opti-MEM I (1X) + GlutaMAX (Gibco). Thirty μl of each dilution was added to monolayers of $2 \times 10^4$ VeroE6, VeroE6-TMPRSS2 or $8 \times 10^4$ Calu-3 cells in the same medium in a 96-well plate. Titrations were performed in triplicate. Plates were incubated

at 37°C overnight and then scanned on the Amersham Typhoon Biomolecular Imager (channel Cy2; resolution 10 µm; GE Healthcare). Entry routes were determined by pre-treating monolayers of VeroE6 or VeroE6-TMPRSS2 cells with a concentration range of camostat mesylate (Sigma) or E64D (MedChemExpress) diluted in Opti-MEM I (1X) + GlutaMAX (Gibco) for 2 hr prior to infection with 1 $\times$ $10^3$ pseudovirus. Plates were incubated at 37°C overnight and then scanned on the Amersham Typhoon Biomolecular Imager (channel Cy2; resolution 10 mm; GE Healthcare). All pseudovirus experiments were quantified using ImageQuant TL 8.2 image analysis software (GE Healthcare).

## Pseudovirus concentration

Pseudoviruses were concentrated as described before (*Mykytyn et al., 2021*) on a 10% sucrose cushion (10% sucrose, 15 mM Tris–HCl, 100 mM NaCl, 0.5 mM EDTA) for 1.5 hr at 20,000 x g at 4°C. Supernatant was decanted and pseudoviruses resuspended in Opti-MEM I (1X) + GlutaMAX (Gibco) to achieve 100-fold concentration.

## Immunoblotting

Concentrated pseudovirus stocks were diluted to a final concentration of 1x Laemmli loading buffer (Bio-Rad) containing 5% 2-mercaptoethanol. Authentic viruses were diluted to a final concentration of 2x Laemmli loading buffer containing 5% 2-mercaptoethanol. All samples were boiled for 30 min at 95°C. Samples were used for SDS-PAGE analysis using precast 10% TGX gels (Bio-Rad). Gels were run in tris-glycine SDS (TGS) buffer at 50V for 30 min and subsequently at 120V for 90 min. Transfer was performed at 300mA for 55 min onto 0.45 µm Immobilon-FL PVDF membranes in TGS containing 20% methanol. Spike was stained using polyclonal rabbit-anti-SARS-CoV S1 (1:1000, Sino Biological), mouse-anti-SARS-CoV-2 S2 (1:1000, Genetex), SARS-CoV-2 nucleoprotein was stained using rabbit-anti-SARS-CoV NP (1:1000, Sino Biological) and VSV nucleoprotein was stained using mouse-anti-VSV-N (1:1000, Absolute Antibody) followed by infrared-labelled secondary antibodies (1:20,000; Licor). Western blots were scanned on an Odyssey CLx and analyzed using Image Studio Lite Ver 5.2 software.

## GFP-complementation fusion assay

Fusion assays were performed as described before (*Mykytyn et al., 2021*). Briefly, HEK-293T cells were transfected with 1.5 µg pGAGGS-spike (all coronavirus S variants described above) DNA and pGAGGS-β-Actin-P2A-7xGFP11-BFP DNA or empty vector DNA with PEI in a ratio of 1:3 (DNA: PEI). Transfected HEK-293T cells were incubated overnight at 37°C 5% $CO_2$, resuspended in PBS and added to GFP1-10 expressing VeroE6, VeroE6-TMPRSS2 and Calu-3 cells in Opti-MEM I (1X) + GlutaMAX at a ratio of 1:80 (HEK-293T cells: GFP1-10 expressing cells). Fusion events were quantified by detecting GFP+ pixels after 18 hr incubation at 37°C 5% $CO_2$ using Amersham Typhoon Biomolecular Imager (channel Cy2; resolution 10 µm; GE Healthcare). Data was analyzed using the ImageQuant TL 8.2 image analysis software (GE Healthcare) by calculating the sum of all GFP+ pixels per well.

## Plaque assay

Virus stock were diluted in 10-fold serial dilutions in 2 ml Opti-MEM I (1X) + GlutaMAX (Gibco). One ml of each dilution was added to monolayers of $2 \times 10^6$ VeroE6 cells in the same medium in a six-well plate. Cells were incubated at 37°C for 1 hr and then overlaid with 1.2% Avicel (FMC biopolymers) in Opti-MEM I (1X) + GlutaMAX (Gibco) for 72 hr. Next, they were washed once in PBS, fixed in formalin, permeabilized in 70% ethanol and washed in PBS again. Cells were blocked in 3% BSA (bovine serum albumin; Sigma) in PBS, stained with mouse anti-nucleocapsid (Sino biological; 1:1000) in PBS containing 0.1% BSA, washed three times in PBS, then stained with goat anti-mouse Alexa Fluor 488 (Invitrogen; 1:2000) in PBS containing 0.1% BSA and then washed three times in PBS. All staining steps were performed at room temperature for one hour. Plates were scanned on the Amersham Typhoon Biomolecular Imager (channel Cy2; resolution 10 µm; GE Healthcare).

## Sanger sequencing

To sequence spike gene fragments, RNA was extracted as described above and used for cDNA synthesis using Superscript IV (Invitrogen), according to the manufacturer's instructions. PCR was

performed using PfuUltra II Fusion HS DNA Polymerase (Agilent Technologies) and primers 5'-TGA-CACTACTGATGCTGTCCGTG-3' and 5'-GATGGATCTGGTAATATTTGTG-3' under the following conditions: initial denaturation at 95℃ for 3 min, followed by 25 cycles of (95℃ for 20 s, 52℃ for 20 s, and 72℃ for 60 s), and a final extension at 72℃ for 10 min. The amplicons were purified (Qiagen PCR purification kit, according to manufacturer) and sequenced with the forward primer using the BigDye Terminator v3.1 Cycle Sequencing Kit and an ABI PRISM 3100 genetic analyzer (Applied Biosystems). The obtained sequences were assembled and aligned using Benchling (MAFFT algorithm).

## Fixed immunofluorescence microscopy and immunohistochemistry

Transwell inserts were fixed in formalin, permeabilized in 0.1% Triton X-100, and blocked for 60 min in 10% normal goat serum in PBS (blocking buffer). Cells were incubated with primary antibodies overnight at 4℃ in blocking buffer, washed twice with PBS, incubated with corresponding secondary antibodies Alexa488-, 594-conjugated secondary antibodies (1:400; Invitrogen) in blocking buffer for 2 hr at room temperature, washed two times with PBS, incubated for 10 min with Hoechst, washed twice with PBS, and mounted in Prolong Antifade (Invitrogen) mounting medium. SARS-CoV-2 was stained with rabbit-anti-SARS-CoV nucleoprotein (40143-T62, 1:1000, Sino biological). Ciliated cells were stained with mouse-anti-AcTub (sc-23950 AF488, 1:100, Santa Cruz Biotechnology). For TMPRSS2 stainings, formalin-fixed inserts were paraffin-embedded, sectioned, and deparaffinized as described before prior to staining (*Rockx et al., 2020*). Samples were imaged on a LSM700 confocal microscope using ZEN software (Zeiss). Immunohistochemistry was performed as described previously (*Rockx et al., 2020*) on formalin fixed, paraffin-embedded Transwell inserts. TMPRSS2 was stained using mouse-anti-TMPRSS2 (sc-515727, 1:200, Santa Cruz Biotechnology), and visualized with goat-anti-mouse (PO260, 1:100, Dako) horseradish peroxidase labeled secondary antibody, respectively. Samples were counterstained using haematoxylin.

## Illumina sequencing

For deep- sequencing, RNA was extracted as described above and subsequently cDNA was generated using ProtoscriptII reverse transcriptase enzyme (New England BiotechnologieBioLabs) according to the manufacturer's protocol. A SARS-CoV-2 specific multiplex PCR was performed as recently described (*Oude Munnink et al., 2020*). In short, primers for 86 overlapping amplicons spanning the entire genome were designed using primal scheme (http://primal.zibraproject.org/). The amplicon length was set to 500 bp with 75 bp overlap between the different amplicons. Amplicons were purified with 0.8x AMPure XP beads (Beckman Coulter) and 100 ng of DNA was converted into paired-end Illumina sequencing libraries using KAPA HyperPlus library preparation kit (Roche) with the KAPA unique dual-indexed adapters (Roche), following the manufacturer's recommendations. The barcode-labeled samples were pooled and analyzed on an Illumina sequencer V3 MiSeq flowcell (2 × 300 cycles).

## Sequencing data analysis

Adapters from the paired-end sequencing reads were trimmed using cutadapt (https://doi.org/10.14806/ej.17.1.200) via: cutadapt -B AGATCGGAAGAGCGTCGTGTAGGGAAAGAGTG -b AGATCGGAAGAGCACACGTCTGAACTCCAGTCAC `–interleaved` `–minimum-length 50`. The trimmed reads were aligned to the genome of Bavpat-1 with Bowtie2 (*Langmead and Salzberg, 2012*) using parameters: `–no-discordant` `–dovetail` `–no-mixed` `–maxins` 2000. Primer sequences were trimmed off from the alignments by soft-clipping the leftmost 33 bases from each sequencing reads using BamUtil (Jun, Wing, *Jun et al., 2015*) via: trimbam {bam_file} - -L 30 R 0 `–clip`. Variants calling was done using VarScan2 (*Koboldt et al., 2012*) and SAMtools (*Li et al., 2009*) via: samtools mpileup `–excl-flags 2048 –excl-flags 256 –fasta-ref {REFERENCE_FAASTA} –max-depth 50000 –min-MQ 30 –min-BQ 30 {BAM_FILE} | varscan pileup2cns –min-coverage 10 –min-reads2 2 –min-var-freq 0.01 –min-freq-for-hom 0.75 –p-value 0.05 –variants 1 > {snp_file}`. Sequence logo were generated with logomaker (*Tareen and Kinney, 2020*) using a custom python script. Plotting of mutation frequencies was done using R and ggplot2 (*Hadley, 2016*). All scripts used for data processing are deposited in GitHub: https://github.com/nicwulab/SARS-CoV-2_in_vitro_adaptation (*Wu et al., 2021*).

Raw sequencing data has been submitted to the NIH Short Read Archive under accession number: BioProject PRJNA694097.

## Statistics

Statistical analysis was performed with the GraphPad Prism 8 and 9 software using an ANOVA or two-way ANOVA followed by a Bonferroni multiple-comparison test.

## Acknowledgements

This work was supported by the Netherlands Organization for Health Research and Development (ZONMW) grant agreement 10150062010008 to BLH and co-funded by the PPP Allowance (grant agreement LSHM19136) made available by Health Holland, Top Sector Life Sciences and Health, to stimulate public-private partnerships. The present manuscript was part of the research program of the Netherlands Centre for One Health. The funders had no role in study design, data collection and interpretation, or the decision to submit the work for publication.

## Additional information

### Funding

| Funder | Grant reference number | Author |
| --- | --- | --- |
| ZonMw | 10150062010008 | Bart L Haagmans |
| PPP allowance | LSHM19136 | Bart L Haagmans |

The funders had no role in study design, data collection and interpretation, or the decision to submit the work for publication.

### Author contributions

Mart M Lamers, Anna Z Mykytyn, Tim I Breugem, Conceptualization, Validation, Investigation, Visualization, Methodology, Writing - original draft, Writing - review and editing; Yiquan Wang, Douglas C Wu, Data curation, Software, Formal analysis, Visualization, Methodology; Samra Riesebosch, Validation, Investigation, Visualization, Methodology; Petra B van den Doel, Investigation, Writing - original draft; Debby Schipper, Investigation; Theo Bestebroer, Investigation, Methodology, Writing - original draft; Nicholas C Wu, Data curation, Software, Formal analysis, Supervision, Visualization, Writing - review and editing; Bart L Haagmans, Conceptualization, Supervision, Funding acquisition, Writing - review and editing

### Author ORCIDs

Mart M Lamers https://orcid.org/0000-0002-1431-4022
Anna Z Mykytyn http://orcid.org/0000-0001-7188-6871
Tim I Breugem http://orcid.org/0000-0002-5558-7043
Yiquan Wang http://orcid.org/0000-0002-1954-9808
Douglas C Wu https://orcid.org/0000-0001-6179-3110
Bart L Haagmans https://orcid.org/0000-0001-6221-2015

### Decision letter and Author response

Decision letter https://doi.org/10.7554/eLife.66815.sa1
Author response https://doi.org/10.7554/eLife.66815.sa2

## Additional files

### Supplementary files

• Transparent reporting form

## Data availability

All scripts used for data processing are deposited in GitHub: https://github.com/wchnicholas/SARS_CoV2_mutation (copy archived at https://archive.softwareheritage.org/swh:1:rev:0e62dbb51f3a470936981d205320ac4ac863e3c3). Raw sequencing data are available at the NIH Short Read Archive under accession number: BioProject PRJNA694097.

The following dataset was generated:

| Author(s) | Year | Dataset title | Dataset URL | Database and Identifier |
|---|---|---|---|---|
| Lamers MM, Mykytyn AZ, Breugem TI, Wang Y, Wu DC, Riesebosch S, van den Doel PB, Schipper D, Bestebroer T, Wu NC, Haagmans BL | 2021 | Human airway cells prevent SARS-CoV-2 multibasic cleavage site cell culture adaptation | https://www.ncbi.nlm.nih.gov/bioproject/PRJNA694097 | NCBI BioProject, PRJNA694097 |

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
