## [Decision Letter]

**Acceptance summary:**

As with many viruses, SARS-CoV-2 grown in transformed cell lines undergoes adaptive changes. This manuscript shows that growing SARS-CoV-2 on relevant respiratory epithelial cell lines or differentiation stem cell cultures prevents the emergence of mutations in the spike (S) protein than impact on cell tropism and viral entry mechanisms. The paper presents a compelling case for culturing primary virus isolates on relevant cell systems and for the use of deep sequencing to analyze viral populations.

**Decision letter after peer review:**

Thank you for submitting your article "Human airway cells prevent SARS-CoV-2 multibasic cleavage site cell culture adaptation" for consideration by *eLife*. Your article has been reviewed by 3 peer reviewers, one of whom is a member of our Board of Reviewing Editors, and the evaluation has been overseen by Jos van der Meer as the Senior Editor. The following individual involved in review of your submission has agreed to reveal their identity: Benjamin G Hale (Reviewer #3).

Essential Revisions:

1. Please comment on any other potential adaptations in regions other than S and the S MBCS. Identifying other potential adaptations is likely to be of value to other researchers.

2. Please mention the results characterising the S686G mutant (and perhaps other mutants) in the abstract to make researchers aware that it is not just deletions in the MBCS that may be adaptations to Vero cells.

3. Please discuss similar lab adaptations in Coronaviruses e.g. 229E (Bertram et al., JVI, 2013; Shirato et al., JVI, 2016) and perhaps mention other virus adaptations to tissue culture lines (e.g. HIV) to broaden awareness of this general problem beyond SARS-CoV-2.*Reviewer #1 (Recommendations for the authors):*

As viruses cultured in Vero-TMPRSS2 cells show a reduced frequency of MBCS mutations, at least over one passage (Figure 6), are these cells a viable alternative for growing virus at titres higher than can be obtained from CaLu-3 or stem-cell derived cultures.

Were the virus stocks from differentiated airway cultures from single or multiple Transwells. Approximately how many cells do these cultures contain and is there significant loss of infectivity during the preparation of the virus (e.g., through centrifugation, media exchange, etc)?

Scale bars are needed on Fig, 1G, 7C and 8C.

*Reviewer #2 (Recommendations for the authors):*

The data is sound and interesting but I fail to see, beyond a modest expansion of what we already knew (and indeed has been known for some time now from many papers by other groups and this group), what the authors are trying to achieve or why it merits anything beyond a short note.

*Reviewer #3 (Recommendations for the authors):*

I found the manuscript clear, convincing and important.

---

## [Author Response]

Essential Revisions:1. Please comment on any other potential adaptations in regions other than S and the S MBCS. Identifying other potential adaptations is likely to be of value to other researchers.

We have not seen any other mutations in the rest of the genome that become dominant variants during passaging in VeroE6, Calu-3 cells and airway organoids, but we note that we cannot exclude that combinations of minor variants make up adapted viruses.

This is now discussed in lines 310-314: “The human airway organoid model for SARS-CoV-2 propagation established in this study (Figure 8) allows high titer SARS-CoV-2 production and was most successful in removing MBCS mutations. We have not observed any major variants associated with virus propagation in this system, but we cannot exclude that some minor variants could be culture adaptations ”.

2. Please mention the results characterising the S686G mutant (and perhaps other mutants) in the abstract to make researchers aware that it is not just deletions in the MBCS that may be adaptations to Vero cells.

We have added our results on the S686G mutation in the abstract: “Here, we report that propagating SARS-CoV-2 on the human airway cell line Calu-3 – that expresses serine proteases – prevents cell culture adaptations in the MBCS and directly adjacent to the MBCS (S686G).”.

3. Please discuss similar lab adaptations in Coronaviruses e.g. 229E (Bertram et al., JVI, 2013; Shirato et al., JVI, 2016) and perhaps mention other virus adaptations to tissue culture lines (e.g. HIV) to broaden awareness of this general problem beyond SARS-CoV-2.

We have added this point in our Discussion section: “Cell culture adaptations promoting cathepsin-mediated entry were also observed for the human coronavirus 229E and adapted viruses showed a reduced ability to replicate in differentiated airway epithelial cells (Bertram et al., 2013; Shirato et al., 2016)”.

Reviewer #1 (Recommendations for the authors):As viruses cultured in Vero-TMPRSS2 cells show a reduced frequency of MBCS mutations, at least over one passage (Figure 6), are these cells a viable alternative for growing virus at titres higher than can be obtained from CaLu-3 or stem-cell derived cultures.

Indeed, virus grown in VeroE6-TMPRSS2 cells showed a reduced frequency of MBCS mutations, indicating that it could be used as an alternative to Calu-3 or organoid-derived cultures. Prolonged passaging in these cells is expected to further decrease the frequency of MBCS mutations, but other Vero cell adaptations not necessarily related to serine protease expression were recently described (Pohl et al., 2021) and could still occur.

Were the virus stocks from differentiated airway cultures from single or multiple Transwells. Approximately how many cells do these cultures contain and is there significant loss of infectivity during the preparation of the virus (e.g., through centrifugation, media exchange, etc)?

As explained in the methods section (line 389), the organoid-produced stocks were produced on twelve 12mm well inserts. These cultures contain approximately 500,000 cells per insert. We did not compare the loss of infectivity due to centrifugation and medium exchange but this appears to be minimal as the observed titers of the organoid-produced stocks are similar to the titers observed in experiments where supernatants were not purified (Lamers et al., 2020 EMBO).

Scale bars are needed on Fig, 1G, 7C and 8C.

Scale bars have now been added.